# Ten Fast Transfer Learning Models for Carotid Ultrasound Plaque Tissue Characterization in Augmentation Framework Embedded with Heatmaps for Stroke Risk Stratification

**DOI:** 10.3390/diagnostics11112109

**Published:** 2021-11-15

**Authors:** Skandha S. Sanagala, Andrew Nicolaides, Suneet K. Gupta, Vijaya K. Koppula, Luca Saba, Sushant Agarwal, Amer M. Johri, Manudeep S. Kalra, Jasjit S. Suri

**Affiliations:** 1CSE Department, CMR College of Engineering & Technology, Hyderabad 501401, TS, India; sivaskandha@cmrcet.ac.in (S.S.S.); hodcse@cmrcet.ac.in (V.K.K.); 2CSE Department, Bennett University, Greater Noida 203206, UP, India; suneet.gupta@bennett.edu.in; 3Vascular Screening and Diagnostic Centre, University of Nicosia, Nicosia 1700, Cyprus; anicolaides1@gmail.com; 4Department of Radiology, Azienda Ospedaliero Universitaria (A.O.U.), 10015 Cagliari, Italy; lucasabamd@gmail.com; 5Global Biomedical Technologies, Roseville, CA 95661, USA; sushant.ag09@gmail.com; 6Division of Cardiology, Queen’s University, Kingston, ON K7L 3N6, Canada; johria@queensu.ca; 7Department of Radiology, Massachusetts General Hospital, 55 Fruit Street, Boston, MA 02114, USA; mkalra@mgh.harvard.edu; 8Stroke Diagnostic and Monitoring Division, AtheroPoint™ LLC, Roseville, CA 95661, USA

**Keywords:** stroke, carotid plaque characterization, symptomatic vs. asymptomatic, artificial intelligence, transfer learning, heatmaps

## Abstract

*Background and Purpose:* Only 1–2% of the internal carotid artery asymptomatic plaques are unstable as a result of >80% stenosis. Thus, unnecessary efforts can be saved if these plaques can be characterized and classified into symptomatic and asymptomatic using non-invasive B-mode ultrasound. Earlier plaque tissue characterization (PTC) methods were machine learning (ML)-based, which used hand-crafted features that yielded lower accuracy and unreliability. The proposed study shows the role of transfer learning (TL)-based deep learning models for PTC. *Methods:* As pertained weights were used in the supercomputer framework, we hypothesize that transfer learning (TL) provides improved performance compared with deep learning. We applied 11 kinds of artificial intelligence (AI) models, 10 of them were augmented and optimized using TL approaches—a class of Atheromatic™ 2.0 _TL_ (AtheroPoint™, Roseville, CA, USA) that consisted of (**i–ii**) Visual Geometric Group-16, 19 (VGG16, 19); (**iii**) Inception V3 (IV3); (**iv–v**) DenseNet121, 169; (**vi**) XceptionNet; (**vii**) ResNet50; (**viii**) MobileNet; (**ix**) AlexNet; (**x**) SqueezeNet; and one DL-based (**xi**) SuriNet-derived from UNet. We benchmark 11 AI models against our earlier deep convolutional neural network (DCNN) model. *Results:* The best performing TL was MobileNet, with accuracy and area-under-the-curve (AUC) pairs of 96.10 ± 3% and 0.961 (*p* < 0.0001), respectively. In DL, DCNN was comparable to SuriNet, with an accuracy of 95.66% and 92.7 ± 5.66%, and an AUC of 0.956 (*p* < 0.0001) and 0.927 (*p* < 0.0001), respectively. We validated the performance of the AI architectures with established biomarkers such as greyscale median (GSM), fractal dimension (FD), higher-order spectra (HOS), and visual heatmaps. We benchmarked against previously developed Atheromatic™ 1.0 _ML_ and showed an improvement of **12.9**%. *Conclusions:* TL is a powerful AI tool for PTC into symptomatic and asymptomatic plaques.

## 1. Introduction

Stroke is the third leading cause of mortality in the United States of America (USA) [1]. According to World Health Organization (WHO) statistics, cardiovascular disease (CVD) causes 17.9 million deaths each year [2]. Atherosclerosis disease is the fundamental cause of CVD, which leads to the formation of complex plaques in the arterial walls owing to a sedentary lifestyle over time [3].

Atherosclerotic plaques, particularly in the internal carotid artery (ICA), may rupture and embolize the brain, leading to stroke. However, only a minority of plaques are unstable and rupture, producing an annual stroke rate of 1–2% in asymptomatic patients with >80% stenosis [4]. Thus, operating on all patients with >80% stenosis will result in many unnecessary operations. In addition, the operation is associated with a 3% preoperative stroke rate. Some plaques are unstable owing to a large lipid core, a thin fibrous cap, and a low collagen content (vulnerable). Therefore, they are more likely to rupture by producing symptoms (symptomatic or hyperechoic or unstable plaque). Compared with the more stable ones, they have a smaller lipid core, a thick fibrous cap, and a large amount of collagen, which tend not to produce symptoms (asymptomatic or hypoechoic or stable plaque) [5]. Therefore, it is important to characterize the plaque early, especially when it is becoming symptomatic or likely to be unstable, leading to rupture with subsequent stroke [6,7].

Several imaging modalities exist to image the plaque, such as magnetic resonance imaging (MRI) [8], computed tomography (CT) [9], and ultrasound (US) [10]. Ultrasound offers essential advantages because it is non-invasive, radiation-free, and portable properties [11,12]. In addition, features like compound and harmonic imaging are now available on standard ultrasonic equipment, yielding a resolution of 0.2 mm [12]. However, visual classification of plaques into stable or unstable using ultrasound images is challenging owing to the inter-variability in plaque tissues [13].

Machine learning is a class of artificial intelligence (AI) that has been previously used for ultrasound-based tissue classification in several organs such as the liver [14,15], thyroid [16,17,18], prostate [19,20], ovary [21], skin cancer [22,23,24,25], diabetes [26,27], coronary [28], and carotid atherosclerotic plaque [22,29,30,31,32]. All these methods use a trial-and-error approach for feature extraction, thus these methods are ad hoc and provide variable results [33]. Therefore, there is a clear need to design and develop automated feature extraction approaches to characterize carotid atherosclerotic plaque into symptomatic and asymptomatic types.

Deep learning (DL) is a subset of AI that has revolutionized image classification methods [34,35,36]. Among all the different DL techniques available, transfer learning (TL) solves the high-performance computational challenges required for images rich with data [37,38,39]. In addition to the computational problem, TL reduces the time taken for training the model compared with DL [40]. This saving of time can be crucial for people with a high risk of stroke [41].

Several popular models exist in TL, and each model offers its own merits and demerits. For example, some models are focused on fast optimization, while some aim for hyperparameter reduction. Some others apply the TL paradigm in edge devices, such as NVIDIA Jetson (www.nvidia.com accessed 20 October 2021) or Raspberry Pi (from Rasberry Pi Foundation, UK) [42]. Few applications of TL have been developed in medical imaging such as classification of Wilson disease [43], COVID pneumonia [44,45,46,47], brain tumour [37], and so on, which has shown superior performance over DL. In this study, we choose ten types of TL architectures, where each one of these carries advantages such as (a) intense neural network, (b) modified kernel sizes, (c) solving vanishing gradient problems, and (d) feed-forward nature to the features [48]. Therefore, we hypothesize that the performance of TL is superior or comparable to that of DL.

The architecture of the proposed global AI model is shown in Figure 1. It contains five blocks: (i) image acquisition, (ii) pre-processing, (iii) AI-based models, and (iv–v) performance evaluation and validation. The image acquisition block is used for scanning the internal carotid artery. These scans are normalized and manually delineated in the pre-processing block to obtain the plaque region-of-interest (ROI). As the cohort size was small, we added the augmentation block as part of the pre-processing step. The AI model block helps to determine whether plaques are symptomatic or asymptomatic. This is accomplished by transforming the test plaque image by the trained TL/DL models. In our proposed framework, because there are 11 models, we run each test patient’s plaque using 11 (10 TL + 1 DL) different AI models for predicting 11 kinds of labels. We determine the performance of these 11 architectures, followed by the ranking of their performance.

We proposed an optimized TL model for carotid ultrasound-based plaque tissue classification (Atheromatic™ 2.0 _TL_, AtheroPoint™, Roseville, CA, USA). Because the features using this system are computed using a deep learning paradigm, we hypothesize that the performance of TL is superior and/or comparable to that of DL. Lastly, we have also designed a computer-aided diagnostics (CAD) system for computing heatmaps using an AI-based approach.

## 2. Literature Survey

The existing work on carotid plaque characterization using ultrasound with AI techniques is primarily focused on the machine learning paradigm. A handful of the studies are focused on using DL. Our study is the first of its kind that uses the TL paradigm embedded with heatmaps for PTC. The section briefly presents the works on PTC. Detailed tabulation is described in the discussion section.

Seabra et al. [49] used graph cut techniques for the characterization of 3D ultrasound. It allows for the detection and quantification of the vulnerable plaque. The same set of authors in [50] estimated the volume inside the ROI plaque using the Bayesian technique. They compared the proposed method with a gold standard and achieved better results with greyscale median (GSM) < 32. In [51], they characterized the plaque components such as lipids, fibrotic, and calcified using the Rayleigh mixture model (RMM).

Afonso et al. [52] proposed a CAD tool (AtheroRisk™, AtheroPoint, Roseville, CA, USA) to characterize the plaque echogenicity using an activity index and enhanced activity index (EAI). The authors achieved an area-under-the-curve (AUC) of 64.96%, 73.29%, and 90.57% for the degree of stenosis, activity index, and enhanced activity index, respectively. This AtheroRisk™ CAD system was able to measure the plaque rupture risk. Loizou et al. identified and segmented the carotid plaque in M-mode ultrasound videos (MUVs) using a snake algorithm [53,54,55]. In [56], the authors studied the variations in texture features such as spatial gray level dependence matrices (SGLD) and gray level difference statistic (GLDS) in the MUV framework to classify them using a support vector machine (SVM) classifier. Doonan et al. [57] studied the relationship between textural and echo density features of carotid plaque by applying the principal component analysis (PCA)-based feature selection technique. The authors showed a moderate coefficient of correlation (r) between these two features, which range from 0.211 to 0.641. In addition to the above studies, Acharya et al. [58,59,60], Gastounioti et. al. [61], Skandha et. al. [62], and Saba et. al. [63] also conducted studies in the area of PTC using AI methods. This will be discussed in detail in Section 5, labeled benchmarking.

## 3. Methodology

This section focuses on patient demographics, ultrasound acquisition, pre-processing, and augmentation protocol. We also described all 11 AI architectures, consisting of ten transfer learning architectures and one deep learning architecture labelled as SuriNet. These are then benchmarked against the deep convolution neural network (DCNN).

### 3.1. Patient Demographics

This cohort consisted of 346 patients with a mean age of 69.9 ± 7.8 and 61% male patients having an internal carotid artery (ICA) stenosis of 50% to 99%. The study was approved by the ethical committee of St. Mary’s Hospital, Imperial College, London, UK (in 2000). The cohort consisted of 196 symptomatic and 150 asymptomatic patients. All the symptomatic patients have ipsilateral cerebral hemispheric symptoms (amaurosis fugax) (AF), transient ischemic attacks, and previous history of stroke. Overall, the symptomatic class contained 38 AF, 70 transient ischaemic attack (TIAs), and 88 strokes, totaling 196. All the asymptomatic patients showed no abnormalities during the neurological study. The same cohort was used in our previous studies [29,32,40,58,62,63,64,65].

### 3.2. Ultrasound Data Acquisition and Pre-Processing

All the US scans were acquired using an ATL machine (Model: HDI 3000; Make: Advanced Technology Laboratories, Seattle, WA, USA) in Irvine Laboratory for Cardiovascular Investigation and Research, St. Mary’s Hospital, UK. This scanner was equipped with a linear broadband width 4–7 MHz (multifrequency) transducer with a 20 pixel/mm resolution. We used proprietary software called “PTAS” developed by Icon soft International Ltd., Greenford, London, UK for normalization and plaque ROI delineation, as used in previous studies [29,32,58,62,64,65]. The medical practitioners delineated the plaque region-of-interest (ROI) using the mouse and trackball; these were then saved in a separate file. Full scans and delineated plaques are shown in Figure 2.

### 3.3. Augmentation

Our cohort was unbalanced, consisting of 196 symptomatic and 150 asymptomatic. Therefore, we choose to balance using the augmentation strategy prior to offline training and online predicting processes. We accomplished this by adding 4 symptomatic and 50 asymptomatic augmented images using random linear transformations such as flipping, rotation by 90 degrees, rotation by 270 degrees, and skew operations. This resulted in a balanced cohort, containing 200 images in each class. Further, the database was incremented two to six times, consisting of an equal number of images using linear transformations. This resulted in six folds of the augmented cohort. We represent these folds as Augmented 2× (Aug 2×), Augmented 3× (Aug 3×), Augmented 4× (Aug 4×), Augmented 5× (Aug 5×), and Augmented 6× (Aug 6×). Thus, every fold contained 200 × n images in each class, where n is the augmented fold.

### 3.4. Transfer Learning

The choice of the TL architecture for PTC was motivated by (a) the diversity of the TL models and (b) the depth of the neural network models. Thus, we took two architectures from the VGG group (VGG-16 and 19), two architectures from the DenseNet architectures (DenseNet121 and 169), and two architectures from the ResNet architectures (ResNet50 and 101). All these models had a depth of neural networks extending to 169 layers while ensuring diversity. Note that some of the architectures such as MobileNet and XceptionNet are the most current, state-of-the-art, and popular TL architectures, demonstrating faster optimization (see Figure 3).

#### 3.4.1. VGG-16 and VGG-19

Visual Geometry Group (VGG-16) is a popular pre-trained model developed by Simonyan et al. [66] to increase the neural networks’ depth by adding a number of 3 × 3 convolution filters. The purpose of VGGx is to design a very deep CNN for complex pattern understanding in the input features, typically adapted for object recognition in medical imaging and computer vision. The architecture of the VGG-16 and 19 is shown in Figure 4, where the input block accepts the image of size 224 × 224. VGG-19 is three layers more than VGG-16 (not shown in the figure). Few applications of VGG-16 and 19 can be seen for the classification of Wilson [38] and COVID-19 pneumonia [67,68] disease.

#### 3.4.2. InceptionV3

InceptionV3 (IV3) is version 3 of the inception stage and was first developed by Szegedy et al. [69]. This model was developed to overcome the computational cost and low parameters count. This model can handle big data. Thus, this model has overall high efficiency. Inception V3 achieves accuracy greater than 78.1% when using the ImageNet dataset. The architecture model contains several blocks. The blocks contain convolution and max-pooling layers. In the architecture given in Figure 5, DL1 to DL6 represent the depth wise convolution, C1 represents the initial convolution block, T1 to T3 represent the transition layer, and D1 to D4 represent the batch normalization blocks. In the Inception V3 architecture, each block in the top row represents the repeated process of row 2 and row 3. In row 2, each block represents the repeated process of row 3. Each convolution layer is fused with a 1 × 1 convolution filter with stride 1 and padding 0. First, it increases the feature map (FM) size, then a 3 × 3 convolution layer with stride 1 and padding 1 is added. It reduces the FM depth; the resultant FM and the initial FM are fused together to give each block in row 2.

#### 3.4.3. ResNet

He et al. [70] from Microsoft research proposed ResNet architecture for solving the vanishing gradient problem. It contains residual blocks. Residual blocks contain skip connections. These skip connections skip some layers from training and connect directly to the output. The advantage of these connections is the skipping of layers, so that the model will learn complex patterns. Unlike other TL models, this model is trained on the CIFAR-10 data set. Figure 6 represents the ResNet architecture. In the architecture, two 3 × 3 convolution layers are paired together. The output of these pairs and its input are fused together and fed to next pair. Here, the number of filters is in increasing order from 64 to 512. At the end of the last 3 × 3 convolution layer with 512 filters and an added flatten layer for vectorization of the 2D features, the output is predicted using the softmax activation function.

#### 3.4.4. DenseNet

Huang et al. [48] proposed the DenseNet architecture for solving vanishing gradient problem in deep neural nets. In this model, dense blocks were introduced. It contains a pool of convolution layers with 3 × 3 filters to 1 × 1 filters followed by batch normalization, and every layer uses the “ReLu” activation function. Each of these dense blocks was concatenated with previous block output and input using transition blocks. Each transition block contains a convolution and pooling layer with 2 × 2 to 1 × 1 filters with dropout layers. This concertation of blocks preserves the feature propagation nature. In addition, the author proposed architectures (DenseNet-121, 169, 201, and 264) to increase the dense block. Figure 7 shows the DenseNet architecture.

#### 3.4.5. MobileNet

Howard et al. [42] from Google developed the MobileNet architecture. The main inspiration of MobileNet comes from the IV3 network. It aims to solve resource constraint problems such as working on edge devices like NVIDIA Jetson (www.nvidia.com accessed 20 October 2021) or Rasberry Pi (from Rasberry Pi Foundation, Cambridge, UK). This architecture is a small, low latency, and low power model. This was the first computer vision model developed for TensorFlow for mobile devices. It contains 28 layers and uses the TFlite (database) library. Figure 8 presents the architecture of MobileNet architecture. This model contains bottleneck residual blocks (BRBs), also referred to as inverted residual blocks used for reducing the number of training parameters in the model.

#### 3.4.6. XceptionNet

Chollet et al. [71] from Google proposed modifying IV3 by replacing the inception modules with modified depth-wise separable convolution layers. This architecture contains 36 layers. In comparison with IV3, XceptionNet is lightweight and contains the same number of parameters as IV3. This architecture outperforms InceptinV3 with top-1 accuracy of 0.790 and top-5 accuracy of 0.945. Figure 9 represents the architecture of XceptionNet.

#### 3.4.7. AlexNet

Alex Krizhevsky et al. [72] proposed AlexNet in 2012 for solving complicated ImageNet challenges. It is the first CNN architecture built for solving complex computer vision problems. This architecture achieves a top-5 error rate of 15.3%. This architecture shifts the paradigm of AI entirely. It takes 256 × 256 size image input and contains five convolution layers followed by max-pooling with two fully connected networks. The output layer is the softmax layer. The sample architecture is shown in Figure 10.

#### 3.4.8. SqueezeNet

Landola et al. [73] proposed a 50× times smaller model than the AlexNet architecture. Nevertheless, the authors achieved 82.5% in top-5 accuracy on ImageNet. This model contains a novel “Fire Module”. It contains a 1 × 1 filtered squeeze convolution layer fed to the “Expand Module”, which contains a mix of 1 × 1 to 3 × 3 filters for convolution. The squeeze layer (Fire Module) helps to reduce the number of input channels to 3 × 3. The architecture of the SqueezeNet and Fire Module is shown in Figure 11. In this study, we transferred trained weights to SqueezeNet initial layers and fed our cohort at the end layer.

### 3.5. Deep Learning Architecture: SuriNet

In our study, we benchmarked TL architectures with two DL architectures. One is conventional CNN and the other is SuriNet architecture. Although the UNet network is very popular for segmentation in medical image analysis, we used a modified UNet architecture called SuriNet for classification purposes. In the proposed SuriNet architecture, we used separable convolution neural networks to reduce the overfitting and the number of parameters required for training. Figure 12 shows the SuriNet architecture. Table 1 gives the detailed number of training parameters for SuriNet.

### 3.6. Experimental Protocol

Our study used 12 AI models (10 TL and 2 DL) with six augmentation folds and 1000 epochs using the K10 cross-validation protocol. It totals to ~720,000 (720 K) runs for finding the optimization point of each AI model. The mean accuracy of each model is calculated using the following section.

#### 3.6.1. Accuracy Bar Charts for Each Cohort Corresponding to All AI Models

If η(m,k) represents the accuracy of an AI model “m” using cross-validation combination “k” out of total combinations K, then the mean accuracy for all the combinations for the model “m”, represented by η(m) can be mathematically given by Equation (1). Note that we considered K10 protocol in our paradigm, so K = K10 = 10.
(1)$$\bar{\eta }\left(m\right)=\frac{1}{K}\overset{K}{\underset{k=1}{\sum }}\eta \left(m,k\right)\;$$

#### 3.6.2. Performance Analysis and Visualization of SuriNet

The objective of this experiment was to evaluate the performance of SuriNet using Equation (1). In addition, SuriNet is based on the DL model. It is end-to-end trained on the target labels. So, we can visualize the intermediate layers’ feature maps of symptomatic and asymptomatic plaques. In this regard, we considered the optimized augmentation fold out of 10 combinations as the combination with the best performance for the visualization of the filters.

## 4. Results

This section discusses *three* sets of experimentations for comparison of TL versus DL to prove the hypothesis. The first experiment is the 3D optimization of the ten TL architectures by varying the augmentation folds. The second experiment is the 3D optimization of the SuriNet architecture by varying the same fold. The third experiment is the benchmarking TL architectures with SuriNet and CNN by calculating the AUC.

### 4.1. 3D Optimization of TL Architectures and Benchmarking against CNN

In this experiment, we used all the TL architectures for finding the optimized TL by varying the augmentation folds. There are 10 TL architectures, 6 augmentation folds, K10 cross-validation protocol, and 1000 epochs. The model is trained by empirically selecting each model’s flatten point at a loss versus accuracy, thus there were 12 × 6 × 10 × 1000 ~720 K runs. We used a total of 720,000 runs to obtain the optimization point. This is a reasonably large number of computations and needs high computation power. Thus, we used the Nvidia DGX V100 supercomputer at Bennett University, Gr. Noida. Figure 13 shows the performance of ten AI architectures, and the red arrow indicates the optimization point for each AI model when ran over six augmentations. The corresponding values are represented in Table 2. Using Equation (1), we calculate the mean accuracy of the AI models.

As seen in Figure 13, MobileNet and DenseNet 169 show better accuracy than other TL architectures. They showed **96.19%** and **95.64**% accuracy, respectively. Aug 2× is the optimization point for both models. Table 3 shows the comparison between ten types of TL, which include VGG16, VGG19, DenseNet121, XceptionNet, MobileNet, AlexNet, InceptionV3, and SqueezeNet, along with seven types of DL. The ten types of TL and seven types of DL include CNN5, CNN7, CNN9, CNN11, CNN13, CNN15, and SuriNet, respectively. Note that CNN5 to CNN15 were taken from our previous study [62], so we have elaborated on the CNN architecture in Appendix A.

In the SuriNet architecture, there are 22 layers, while there is a varying number of layers in the CNN architecture, ranging from 5 to 15. It is important to note that all CNNs except CNN5 have accuracies above 92.27%. The overall mean and standard deviation of the DL accuracies was **90.86 ± 3.15%**. The innovation of the current study was the design and development of TLs. They are benchmarking against DL. In Table 3, the mean and standard deviation of ten TLs was **89.35 ± 2.54%**. Thus, the mean accuracy of TL systems is comparable to the mean accuracy of DL systems and in the range of **~1%**. MobileNet has the highest accuracy among all the TL systems (**96.19**%), while CNN11 has the highest accuracy among all the DL systems (95.66%). Further, it is essential to note that the mean accuracy variations are less than or equal to 3% within the limits of good design and operating conditions (typically, regulatory approved systems have variation of less than 5%).

### 4.2. 3D Optimization of SuriNet

In this set of experiments, we used the popular UNet architecture model for classification. Figure 12 represents the SuriNet architecture inspired by UNet. We optimized SuriNet by varying the augmentation folds. Here, we also used the K10 CV protocol for training and testing. We choose 1000 epochs empirically. Therefore, the total number of runs for optimizing SuriNet is 60,000 (1 SuriNet × 6 Aug folds × 10 combinations × 1000 epochs). We used the same set of hardware resources (used in the previous section) for this experiment. Table 2 represents the average accuracy at the augmentation folds. SuriNet is optimized at Aug 5× with an accuracy of **92.77** percent.

### 4.3. Visualization of the SuriNet

We visualized the intermediate layers of SuriNet to understand the learning ability of the model over CUS. Figure 14 represents the mean visualization of the training samples of symptomatic and asymptomatic classes from all the filters at the end layer before vectorization. The turquoise color represents the learned features, yellow represents the high-level features, and green represents the low-level features.

## 5. Performance Evaluation

This section aims to evaluate the samples required for the study using standard power analysis. As we are using 12 AI models (10 TL, 2 DL), it is necessary to rank the models by considering all the performance parameters for finding the best performing AI model among the 12 AI models. In addition to that, we compared the performance of all 12 AI models with area-under-the-curve (AUC) using the receiver operating characteristic curve (ROC).

### 5.1. Power Analysis

We used a standardized protocol (power analysis) for analyzing the number of samples required at a certain threshold of the error margin. We considered a 95% confidence interval with a 5% margin of error and a data proportion of 0.5. We used Equation (2) below to compute the number of samples.
(2)$$n=\left[{\left(z^{*}\right)}^{2}\times \left(\frac{\overset{\text{\textasciicircum{}}}{p}\left(1-\overset{\text{\textasciicircum{}}}{p}\right)}{{\mathrm{MoE}}^{2}}\right)\right]$$

Here, n is the number of samples (sample size), z* is the z score (1.96) from the z-table, MoE is a margin of error, and $\overset{\text{\textasciicircum{}}}{p}$ represents the data proportion. In our study, we had a total of 2400 images. Using the power analysis, the total samples required for the study was 384. Thus, the number of the sample used in this study was 84% higher than the required samples.

### 5.2. Ranking of AI Models

After obtaining the absolute values of 12 AI models’ performance metrics, we sorted the AI models into increasing order and then compared each value with the highest possible value in the attribute. We considered five marks. If the percentage was more significant than 95%, we considered four marks. If it was greater than 90 and less than 95, we considered three marks. If it was more significant than 80% and less than 90%, we considered two marks. If it was more significant than 75%, we considered one mark. If it was greater than 50% or less than 50%, it was considered as zero. The resultant rank table of the AI models is shown in Table 4. We color-coded each AI model from red to green. Each model is color-coded in this band. If the model performance is low, it is represented as red. If it performs well, it is represented as green. Please see Appendix B for grading scheme.

### 5.3. AUC-ROC Analysis

We computed the area-under-the-curve (AUC) for all the proposed AI models and compared the performance with our previous existing work [62] consisting of a CNN model with an accuracy of 95.66% and AUC of 0.956. Figure 15 represents the ROC comparison of 10 AI methods. Among all the architectures, MobileNet showed the highest AUC value as 0.961 (*p*-value < 0.0001) and better performance than CNN [62].

## 6. Scientific Validation versus Clinical Validation

In this section, we discussed the validation of the hypothesis. Scientific validation was carried out by heatmap analysis using the TL-based “Grad Cam” technique and clinical validation was proved using a correlation analysis of the biomarker with AI.

### 6.1. Scientific Validation Using Heatmaps

We applied a novel visualization technique called gradient weighted class activation *map* (“*Grad Cam*”) for identifying the diseased areas in the plaque cut sections using VGG16 transfer learning architecture. Grad-CAM produces heatmaps based on the weights generated during the training. Here, we take feature maps of the final layer. It gives the essential regions of the target, and heatmaps highlight these regions. Figure 16 and Figure 17 represent the heatmaps of the nine patients of symptomatic and asymptomatic class. The dark red color region represents the diseased region in symptomatic plaque, whereas it represents the higher calcium area in asymptomatic plaque.

The Grad-Cam works on the training weights generated during the training phase. The DL model captures the important regions of the target label. We compared the heatmaps with original images of both symptomatic and asymptomatic images. We observed that heatmaps exhibit a darker region surrounded by grayscale regions. Meanwhile, in asymptomatic regions, DL observes grayscale regions. Figure 17(a1,a2,b1,c1) are the important regions observed by DL of symptomatic images, and Figure 17(d1,e1,e2,e3,f1,f2,f3) are the observed important regions of the asymptomatic images by the DL model. This comparison proves our hypothesis that symptomatic plaques are hypoechoic and dark, and asymptomatic plaques are bright and hyperechoic.

### 6.2. Correlation Analysis

We correlated all the biomarkers for the detection of the risk with AI. Table 5 represents the correlation coefficient of all the biomarkers. Among all the biomarkers, GSM versus FD shows a better *p*-value. We computed the correlation coefficient using MedCalc. We computed the Euclidean distance (ED) between the centers of the two clusters (sym and asym). Table 6 represents the ED between two clusters, symptomatic versus asymptomatic. AI shows constant variation among all the techniques, whereas GSM with FD and higher order spectra (HOS) shows the maximum distance. Figure 18 represents the correlation of AI (SuriNet), GSM, FD, and HOS, and the black dot represents the center of each class. The clusters of symptomatic and asymptomatic are represented with red and violet color, respectively. The black dot represents the center of the cluster and the eclipse on the cluster represents the high-density area. Figure 18b,d,e represent the (a) strong correlation, (c) moderate correlation, and (f) weak correlation between the biomarkers.

## 7. Discussion

The proposed study is the first of its kind to use ten transfer learning models that classify and characterize the symptomatic and asymptomatic carotid plaques. The proposed models, 10 TL and 1 DL (SuriNet), are optimized using augmentation folds with K10 cross-validation protocol. The proposed MobileNet showed an accuracy of **96.19**%, while SuriNet was relatively high, having an accuracy of **92.70**%, and our previous study using CNN [62] showed **95.66**%. Our overall performance analysis showed that TL performance is superior to that of the DL models.

### 7.1. Benchmarking

In this section, we benchmarked the proposed system with the existing techniques [29,58,59,60,61,62,63,74,75,76,77,78,79,80,81,82,83,84]. Table 7 shows the benchmarking table, where the table can be classified into ML-based and DL-based systems for PTC. The table shows columns C1 to C6, where C1 represents the author and the corresponding year, C2 shows the selected features for that study, C3 shows the classifiers used for PTC, C4 displays the dataset size and country, and C5 and C6 give the type of AI model and accuracy along with the AUC. Rows R1 to R17 represent the existing studies on PTC using CUS, while R18 and R19 discuss the proposed studies. In row R1, Christodoulou et al. [76] extracted ten different law texture energy features and fractal dimension features from the CUS and were able to characterize the PTC with diagnostic yield (DY) of 73.1% using SOM and 68.8% using *k-*NN. Mougiakakou et al. (2006) [44] (R2, C1) extracted first-order statistics and the law of texture energy features from 108 US scans. The authors reduced the dimensionality of the extracted features using ANOVA and then fed the resultant features to neural networks with backpropagation and genetic architecture to classify symptomatic versus asymptomatic plaques. The authors achieved an accuracy of 99.18% and 94.48%, respectively. Seabra et al. [74] (R3, C1) extracted echo-morphological and texture features from 146 US scans. Then, they fused those features with clinical information, later used by AdaBoost classifier for classifying symptomatic versus asymptomatic plaques. The authors successfully achieved 99.2% accuracy using leave-one-participant-out (LOPO) cross-validation.

Christodoulou et al. [79] (R4, C1) extracted multiple features such as shape features, morphology features, histogram features, and correlogram features from 274 US scans, which were then used by two sets of classifiers, SOM and *k-*NN. The authors achieved an accuracy of 72.6% and 73.0%, respectively. Acharya et al. [58] (R5, C1) extracted texture-based features from the Cyprus cohort containing 346 carotid ultrasound scans, which were then fed to (a) SVM classifier with RBF kernel and (b) Adaboost classifier. The authors achieved an accuracy of 82.48% and 81.7% with AUC of 0.82 and 0.81, respectively. Kyriacou et al. [80] (R6, C1) developed a CAD system for predicting the period of stroke using binary logistic regression and SVM, which achieved 77%. Acharya et al. [59] (R7, C1) extracted texture-based features from 346 CUS scans and fed them to the SVM classifier, and achieved an accuracy of 83.78%. The same authors in [60] (R8, C1) extracted discrete wavelet transform (DWT) features using the Cyprus cohort of 346 US scans, and fed them to an SVM classifier, achieving an accuracy of 83.78%. Gatounioti et al. [61] (R9, C1) extracted Fisher discriminant ratio features from 56 CUS scans, and fed them to an SVM classifier, achieving an accuracy of 88.08% with an AUC of 0.90. Molinari et al. [84] (R10, C1) used a data mining approach by taking bidimensional empirical mode decomposition and entropy features from 1173 CUS scans and then used an SVM classifier with RBF kernel for classification. The authors achieved an accuracy of 91.43%.

The second set of studies used DL models for PTC. Skandha et al. [62] (R11, C1) extracted automatic features using optimized CNN from augmented 346 patients. The authors achieved an accuracy of 95.66% and an AUC of 0.956 (*p* < 0.0001). The authors successfully characterized the symptomatic versus asymptomatic plaques using mean feature strength, higher-order spectrum, and histogram analysis. Saba et al. [63] (R12, C1) used a randomized augmented cohort generated from 346 patient CUS with 13 layered CNN and achieved an accuracy of 89% with an AUC of 0.9 (*p* < 0.0001).

### 7.2. Comparison of TL Models

TL architectures use the pretrained weights for retraining the model for target label prediction. However, the TL architecture training time depended on the size of the pretrained weights and hardware resources. Various TL models discussed in Table 6 had advantages over the other model, as explained in Table 8 and Table 9.

### 7.3. Advantages of TL Models

TL models’ designs have similarities and differences between them. These are explained in Table 9, along the key findings of every TL model.

### 7.4. GUI Design

AtheroPoint™ developed the Atheromatic™ 2.0 _TL_ system, a computer-aided diagnostic system for stroke risk stratification. Figure 19 represents the screenshot of the CAD system. This CAD system will provide the plaque risk and heatmaps generated by the Grad-Cam with the help of TL/DL models. In the CAD system, the heatmap would be predicted on the test image once the training model is selected.

### 7.5. Strengths/Weakness/Extensions

We evaluated the optimization point of the TL models against various augmentation folds and compared the performance of the TL models against that of the DL models such as SuriNet and CNN. The TL model showed an improvement for symptomatic versus asymptomatic plaque classification accuracy. Furthermore, our Atheromatic™ 2.0 _TL_ system predicts the risk of plaque and vulnerability using the color heatmaps on test scans.

Even though the power sample suggests that we have enough samples for the training, the main limitation of this study was the moderate cohort size. In addition to the cohort size, another limitation of this study is the limited availability of the hardware resources such as supercomputer availability, especially in third-world developing countries.

Our study had a manual delineation of ICA data sets. In future, there could be a need to design an automated ICA segmentation system [85]. Another possibility would be to improve the CNN by an improved DCNN model, where the rectified linear unit (ReLU) activation function was modified, ensuring “differentiable at zero” [38]. There are dense networks such as DenseNet121, DenseNet169, and DenseNet201 that could be tried and compared [39]. Further, one can further combine hybrid deep learning models for PTC [86]. Finally, the proposed AI models can be extended to a big data framework by including other risk factors.

## 8. Conclusions

The proposed study is the first of its kind to characterize and classify the carotid plaque using an optimized transfer learning approach and SuriNet (a class of Atheromatic™ 2.0 _TL_). Eleven AItherop models were implemented, and the best AUC was **0.961 (*p* < 0.0001)** from MobileNet and **0.927 (*p* < 0.0001)** from SuriNet. We validated the performance using grayscale median, fractal dimension, higher-order spectra, and spatial heatmaps. TL showed equal and comparable performance to deep learning. The Atheromatic™ 2.0 _TL_ model showed a performance improvement of **12.9**% over Atheromatic™ 1.0_ML_ (AtheroPoint, Roseville, CA, USA) compared with the previous machine learning-based paradigm. The system was validated with the widely accepted dataset.

## Figures and Tables

**Figure 1 diagnostics-11-02109-f001:**
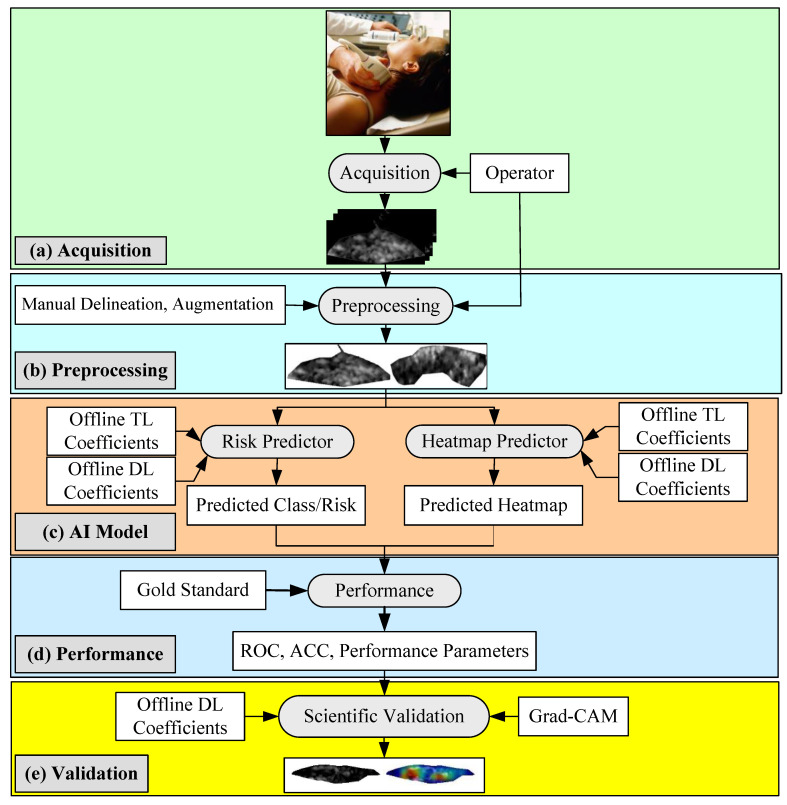
Online AI architecture of the Atheromatic™ 2.0 _TL_ study (TL: transfer learning, DL: deep learning, and Grad-CAM: gradient-weighted class activation mapping).

**Figure 2 diagnostics-11-02109-f002:**
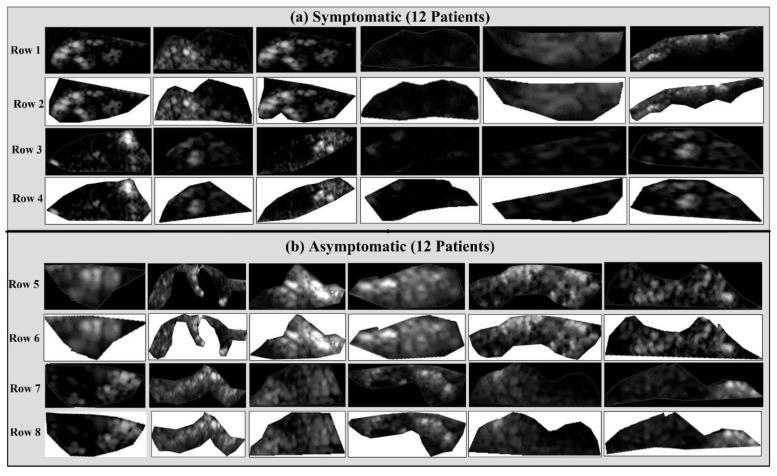
(**a**) Top: symptomatic (row 1 and row 3) and (**b**) down: asymptomatic; (row 5 and row 7): original carotid full scans; row 2, row 4, row 6, and row 8 are the plaque delineated cut sections of (**a**) symptomatic and (**b**) asymptomatic plaques after pre-processing and delineation.

**Figure 3 diagnostics-11-02109-f003:**
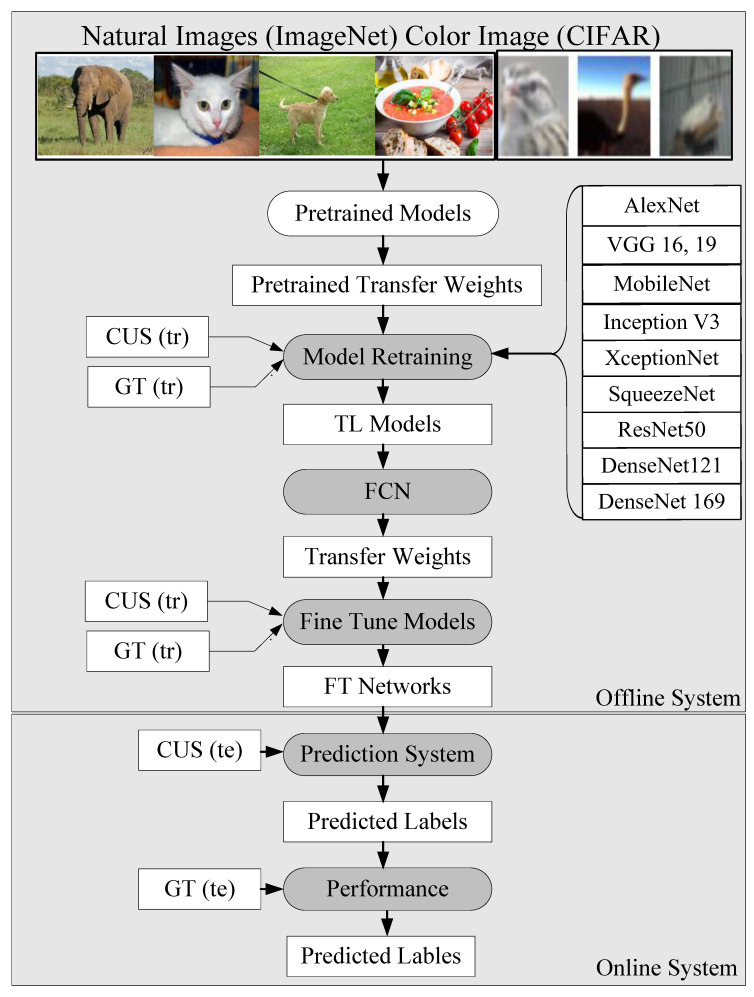
Global TL architecture using 10 different TL models (**i**–**ii**) Visual Geometric Group-16, 19 (VGG16, 19); (**iii**) Inception V3 (IV3); (**iv**–**v**) DenseNet121, 169; (**vi**) XceptionNet; (**vii**) ResNet50; (**viii**) MobileNet; (**ix**) AlexNet; and (**x**) SqueezeNet. Te stands for testing and tr stands for training. FN: fine-tune networks.

**Figure 4 diagnostics-11-02109-f004:**
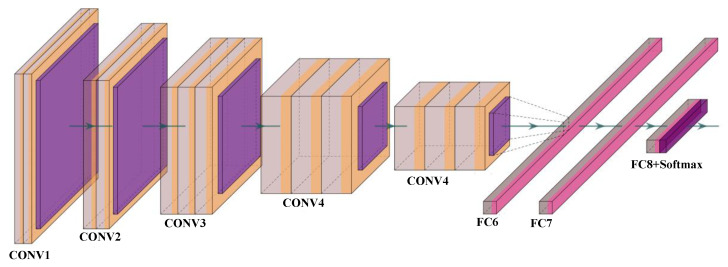
VGG16 and VGG19 architectures; CONV: convolution layer and FC: fully connected network.

**Figure 5 diagnostics-11-02109-f005:**
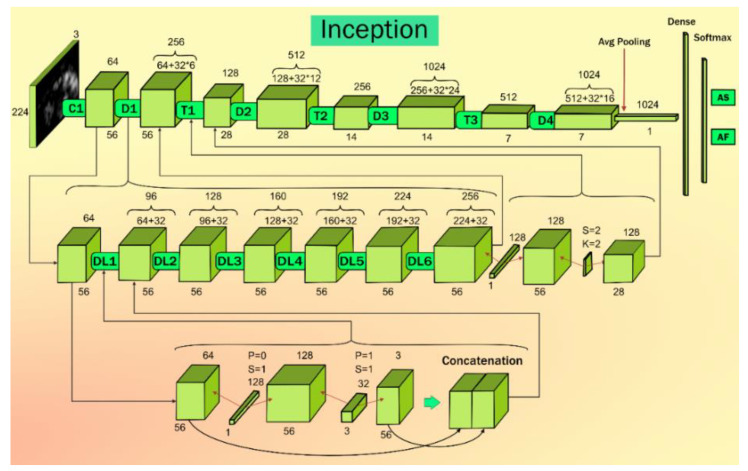
Inception V3 architecture.

**Figure 6 diagnostics-11-02109-f006:**
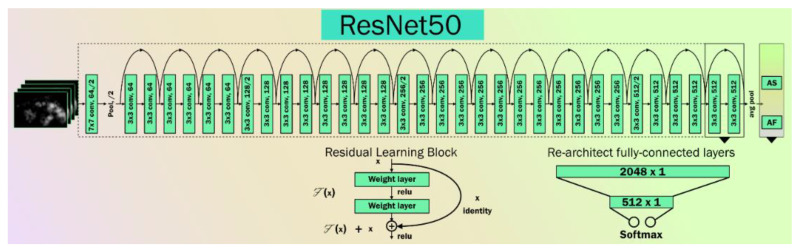
ResNet Architecture.

**Figure 7 diagnostics-11-02109-f007:**
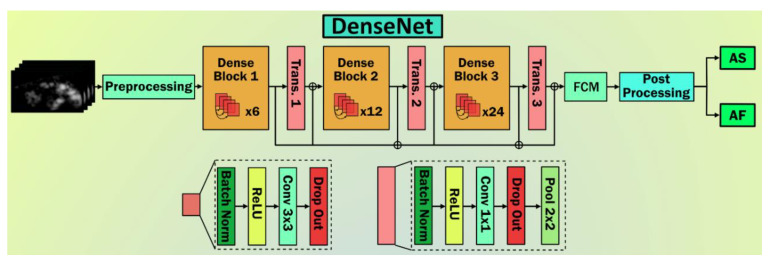
DenseNet architecture with three dense blocks and three transition blocks, followed by the fully connected network. Post processing is represented by softmax.

**Figure 8 diagnostics-11-02109-f008:**
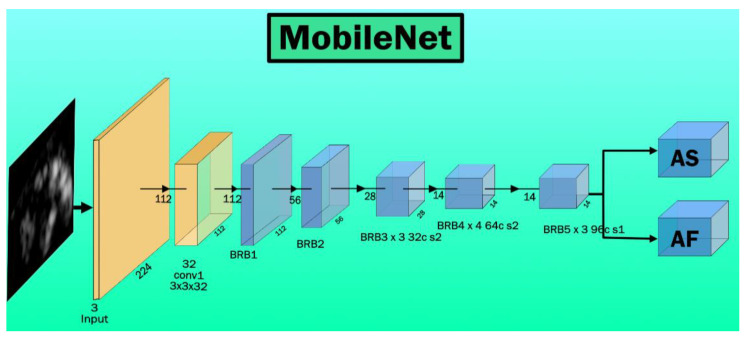
MobileNet Architecture, BRB: bottleneck and residual blocks.

**Figure 9 diagnostics-11-02109-f009:**
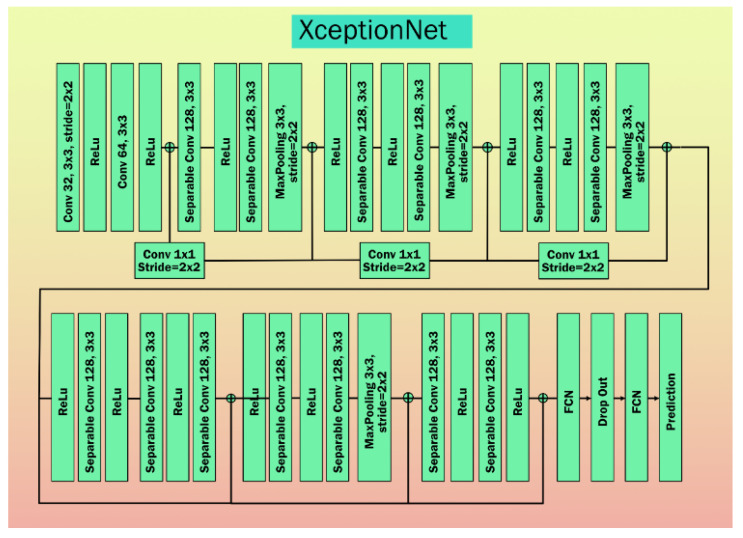
XceptionNet architecture.

**Figure 10 diagnostics-11-02109-f010:**
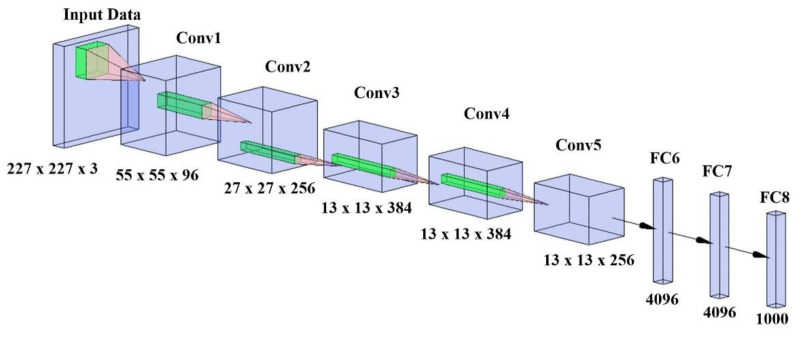
AlexNet architecture.

**Figure 11 diagnostics-11-02109-f011:**
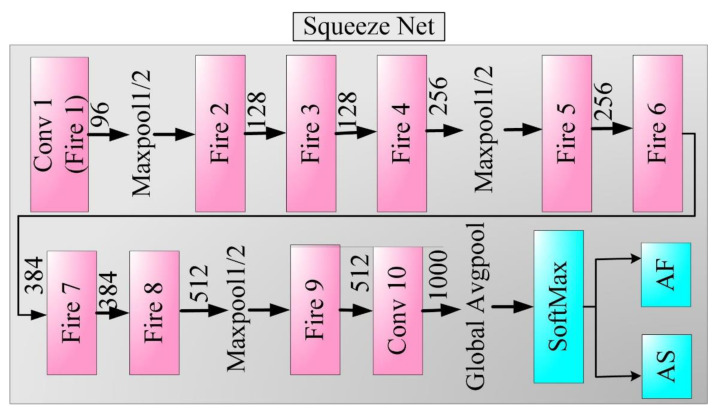
SqueezeNet architecture.

**Figure 12 diagnostics-11-02109-f012:**
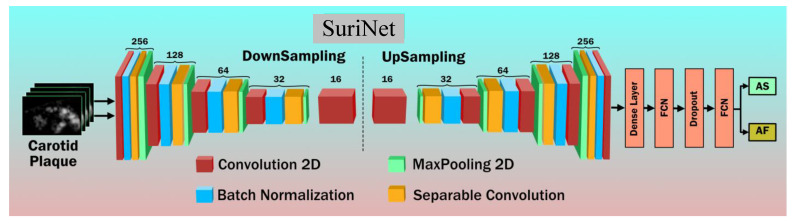
SuriNet architecture.

**Figure 13 diagnostics-11-02109-f013:**
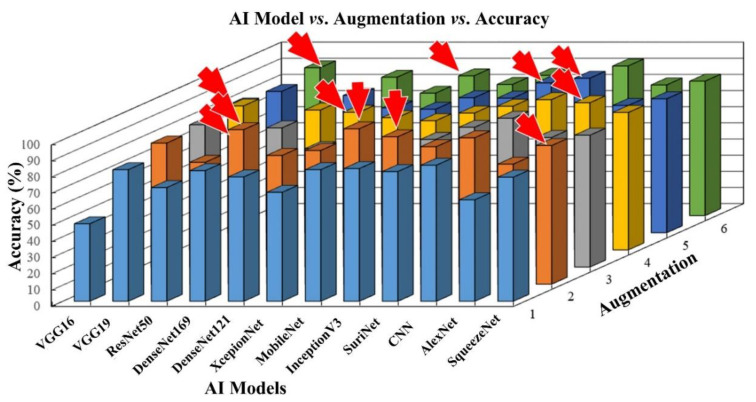
3D bar chart representation of the AI model accuracy vs. augmentation folds, light blue color bar represents the Aug 1×, orange color bar represents the Aug 2×, gray color bar represents Aug 3×, yellow bar represents the Aug 4×, dark blue color represents Aug 5×, green color bar represents Aug 6×, and red arrow represents the optimization point of each classifier.

**Figure 14 diagnostics-11-02109-f014:**
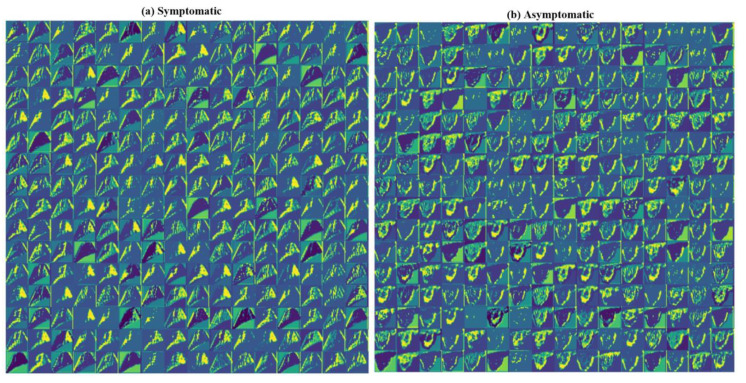
Visualization of the intermediate layers of SuriNet on the (**a**) symptomatic class and (**b**) asymptomatic class.

**Figure 15 diagnostics-11-02109-f015:**
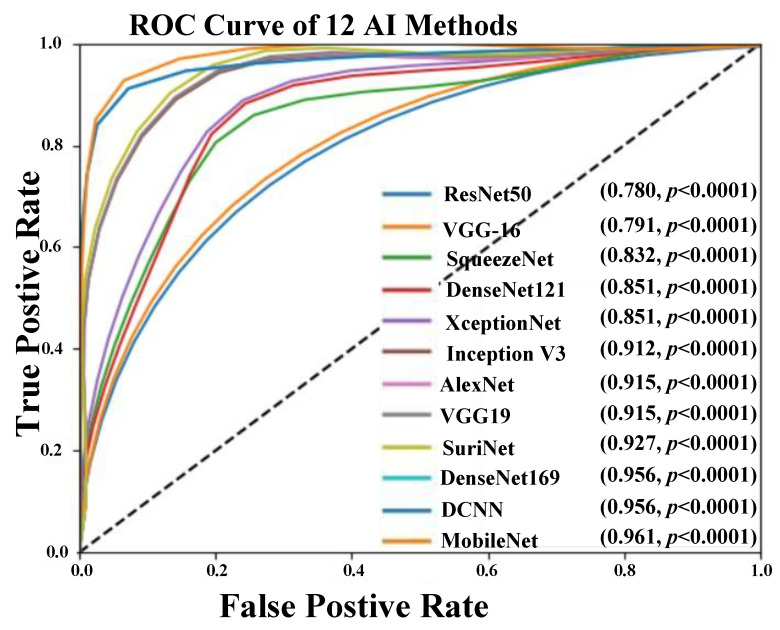
ROC comparison of 12 AI models (10 TL and 2 DL).

**Figure 16 diagnostics-11-02109-f016:**
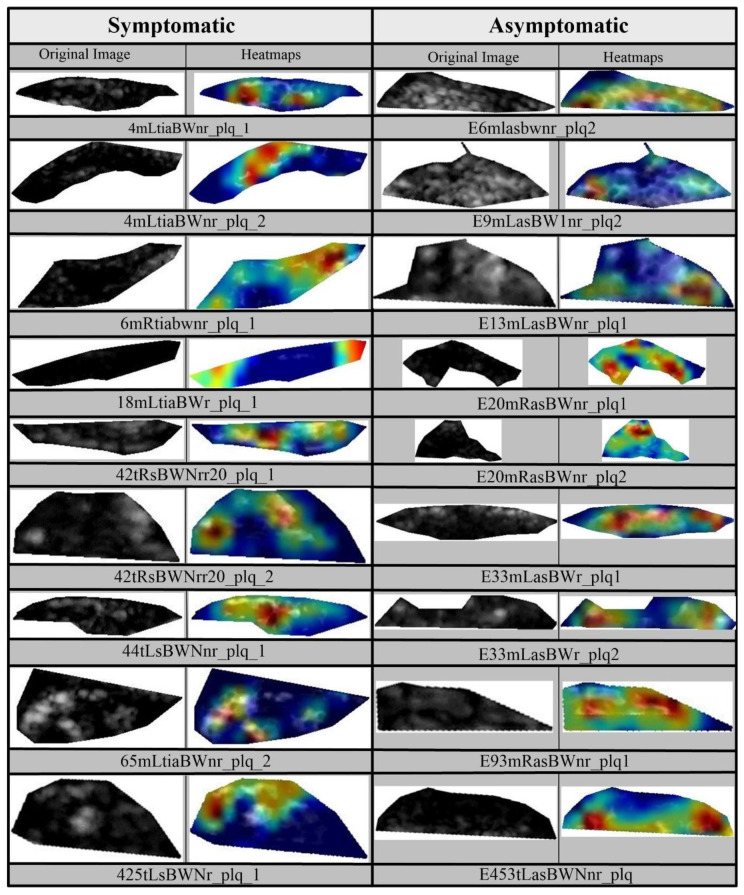
Heat maps of the symptomatic plaque (**left**) and asymptomatic plaque (**right**).

**Figure 17 diagnostics-11-02109-f017:**
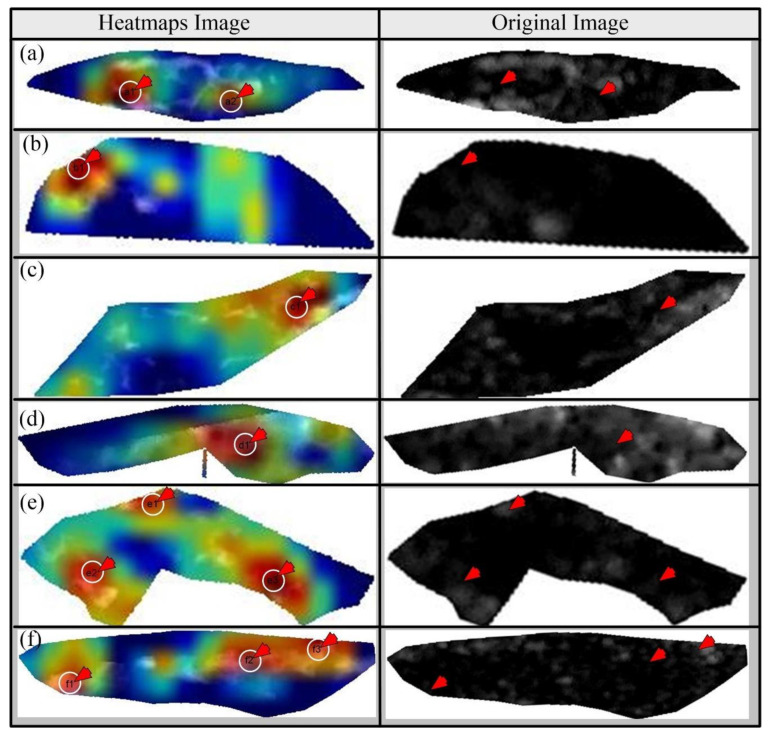
(**a**–**c**) The symptomatic image heatmaps vs. original images; (**d**–**f**) the asymptomatic image heatmaps vs. original images (red color arrow represents the important regions).

**Figure 18 diagnostics-11-02109-f018:**
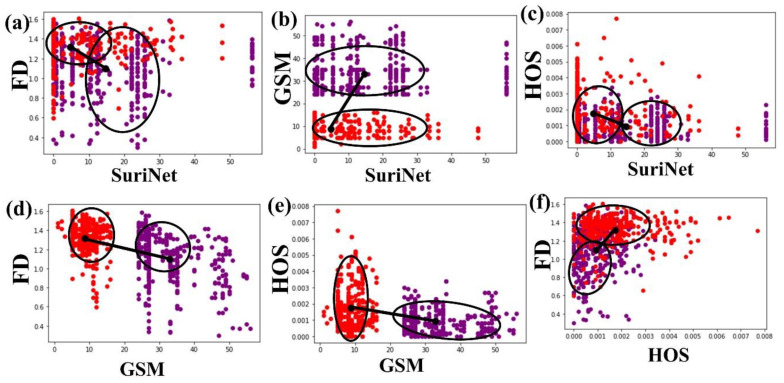
Correlation of AI (SuriNet) and the three biomarkers—FD, GSM, and HOS (**a**) FD vs. SuriNet, (**b**) GSM vs. SuriNet, (**c**) HOS vs. SuriNet, (**d**) FD vs. GSM, (**e**) HOS vs. GSM, and (**f**) FD vs. HOS.

**Figure 19 diagnostics-11-02109-f019:**
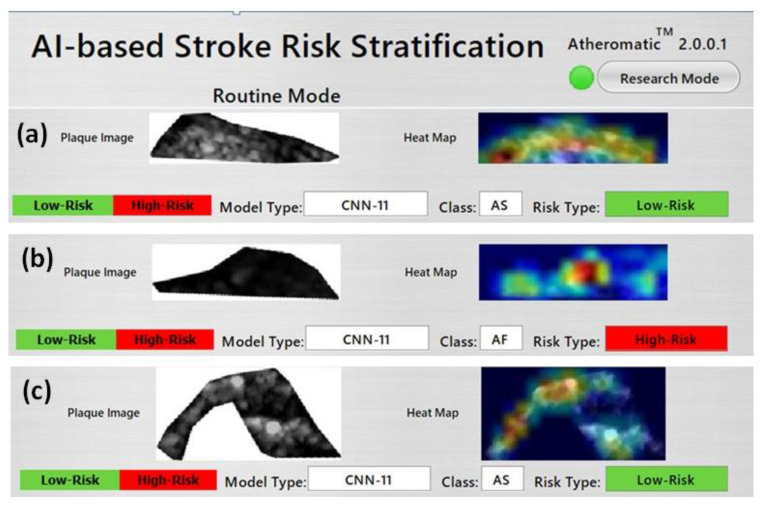
GUI screenshot of the Atheromatic™ 2.0 _TL_ system with three example cases (**a**–**c**).

**Table 1 diagnostics-11-02109-t001:** SuriNet architecture parameters.

Layer Type	Shape	#Param
Convolution 2D	128 × 128 × 32	896
Batch normalization	128 × 128 × 32	128
Separable Convolution 2D	128 × 128 × 64	2400
Batch normalization	128 × 128 × 64	256
MaxPooling 2D	64 × 64 × 64	0
Separable Convolution 2D	64 × 64 × 128	8896
Batch normalization	64 × 64 × 128	512
MaxPooling 2D	32 × 32 × 128	0
Separable Convolution 2D	32 × 32 × 256	34,176
Batch normalization	32 × 32 × 256	1024
MaxPooling 2D	16 × 16 × 256	0
Separable Convolution 2D	16 × 16 × 64	18,752
Batch normalization	16 × 16 × 64	256
MaxPooling 2D	8 × 8 × 64	0
Separable Convolution 2D	8 × 8 × 128	8896
Batch normalization	8 × 8 × 128	512
MaxPooling 2D	4 × 4 × 128	0
Separable Convolution 2D	4 × 4 × 256	34,176
Batch normalization	4 × 4 × 256	1024
MaxPooling 2D	2 × 2 × 256	0
Flatten	1024	0
Dense	1024	1,049,600
Dropout	0.5	0
Dense	512	524,800
Dropout	0.5	0
Dense (softmax)	2	1026
**Total Trainable Parameters**		**1,687,330**

**Table 2 diagnostics-11-02109-t002:** Accuracies of 10 TL and 2 DL models for 6 augmentations. Bold represents the optimization point of each classifier.

AI Model	Balanced	Aug 2×	Aug 3×	Aug 4×	Aug 5×	Aug 6×
VGG16	48	47.5	47.97	66.72	**79.12**	70.87
VGG19	81.5	87.33	88.07	89.08	87.5	**91.56**
ResNet50	70.4	75.4	**78.2**	70.5	68.7	66.5
DenseNet169	80.9	**95.64**	86.14	86.57	85.06	85.66
DenseNet121	76.99	79.69	73.29	**85.17**	77.33	75.81
Xception Net	67.49	82.74	79.99	81.87	76.49	**86.55**
MobileNet	81.49	**96.19**	72.82	79.99	83.59	81.24
InceptionV3	82.18	**91.24**	79	84.69	83.33	86.88
SuriNet	80.32	85.09	86.50	88.93	**92.77**	84.95
CNN [62]	84.24	90.6	92.12	92.99	**95.66**	92.66
AlexNet	62.84	74.29	80.21	**91.09**	78.81	80.91
SqueezeNet	74.65	**83.20**	79.23	83.12	81.33	82.00

**Table 3 diagnostics-11-02109-t003:** TL systems vs. DL systems, background color represents the optimization point.

TL Type	TL Acc. (%)	DL Type	DL Acc. (%)
VGG16	79.12	CNN5	70.32
VGG19	91.56	CNN7	94.24
DenseNet169	95.64	CNN9	95.41
DenseNet121	85.17	CNN11	95.66 *
Xception Net	86.55	CNN13	92.27
MobileNet	96.19 *	CNN15	95.40
InceptionV3	91.24	SuriNet	92.77
AlexNet	91.09		
SqueezeNet	83.20		
ResNet50	78.20		
**Best TL**	**96.19**	**Best DL**	**95.66**
Absolute difference mean TL vs. mean DL			**0.53**

***** Highest accuracy.

**Table 4 diagnostics-11-02109-t004:** Ranking table of the AI models. The background color tells about the intensity of the classifier.

Rank	Model	O	A	F	F1	Se	Sp	DS	D	TT	Me	AUC	AS	%
1	VGG19	5	3	4	5	5	4	5	5	3	1	3	43	78.18
2	MobileNet	2	5	4	3	5	4	1	4	5	5	5	43	78.18
3	CNN11 *	4	5	2	4	5	4	4	5	1	3	5	42	76.36
4	AlexNet	5	4	2	2	2	2	5	3	4	3	3	35	63.60
5	Inception	1	3	5	5	4	5	1	5	1	1	3	34	61.82
6	DenseNet169	1	5	4	3	3	4	1	3	2	3	5	34	61.82
7	XceptionNet	5	3	2	2	3	2	5	0	3	4	3	32	58.18
8	SuriNet	2	3		3	3	4	3	3	3	3	3	30	54.55
9	VGG16	5	1	3	3	3	3	5	1	4	1	1	30	54.55
10	SqueezeNet	2	2	3	3	3	3	4	1	2	3	2	28	50.90
11	DenseNet 121	4	2	2	2	3	2	4	0	2	3	2	26	47.27
12	ResNet50	3	2	2	2	3	2	3	0	1	3	2	23	41.80

O: optimization, A: accuracy, F: false positive rate, F1: F1 score, Se: sensitivity, Sp: specificity, DS: data size, D: DOR, TT: training time, Me: memory, AUC: area-under-the-curve AS: absolute score. * Note that CNN11 (rank 3) was used for benchmarking against other models (1, 2, and 4–12).

**Table 5 diagnostics-11-02109-t005:** Correlation analysis.

Comparison	Symptomatic		Asymptomatic		Abs. Difference
	CC	*p*-Value	CC	*p*-Value	
FD vs. HOS	0.07221	0.0149	0.156	0.0017	1.160366
FD vs. GSM	−0.241	<0.0001	−0.383	<0.0001	0.589212
GSM vs. HOS	0.0725	0.0147	−0.0630	0.0208	1.868966
SuriNet vs. GSM	0.0017	0.009	−0.0437	0.0031	26.70588
SuriNet vs. HOS	−0.0234	0.006	−0.0394	0.0042	0.683761
SuriNet vs. FD	0.0623	0.0021	0.01347	0.0079	0.783788

**Table 6 diagnostics-11-02109-t006:** Euclidean distance between biomarker pairs.

Comparison	Euclidean Distance
SuriNet vs. FD	9.82
SuriNet vs. GSM	9.83
SuriNet vs. HOS	8.83
FD vs. GSM	24.20
GSM vs. HOS	24.19
FD vs. HOS	2.18

**Table 7 diagnostics-11-02109-t007:** Benchmarking table.

SN#	C1	C2	C3	C4	C5	C6
	Authors, Year	Features Selected	Classifier Type	Dataset	AI Type	ACC. (%) AUC (*p*-Value)
R1	Christodoulou et al. (2003) [76]	Texture Features	SOM KNN	230 (-)	ML	73.18, 68.88, 0.753, 0.738
R2	Mougiakakou et al. (2006) [77]	FOS and Texture Features	NN with BP and GA	108 (UK)	ML	99.18, 94.48, 0.918
R3	Seabra et al. 2010 [74]	Five Features	Adaboost using LOPO	146 Patients	ML	99.2
R4	Christodoulou et al. 2010 [79]	Shape Features, Morphology Features, Histogram Features, Correlogram Features	SOM KNN	274 Patients	ML	72.6, 73.0
R5	Acharya et al. (2011) [58]	Texture Features	SVM with RBF Adaboost	346 (Cyprus)	ML	82.48, 81.78, 0.818, 0.810 *p* < 0.0001
R6	Kyriacou et al. 2012 [80]	Texture Features with Second-Order Statistics Spatial Gray Level Dependence Matrices	Probabilistic neural networks and SVM	1121 Patients	ML	77, 76
R7	Acharya et al. (2012) [59]	Texture Features	SVM	346 (Cyprus)	ML	83.8 *p* < 0.0001
R8	Acharya et al., (2012) [60]	DWT Features	SVM	346 (Cyprus)	ML	83.78 *p* < 0.0001
R9	Gastounioti et. al. (2014) [61]	FDR+ Features	SVM	56 US Image	ML	88.08, 0.90
R10	Molinari et al. 2018 [84]	Bidimensional empirical mode decomposition and entropy features	SVM with RBF	1173 Patients	ML	91.43 *p* < 0.0001
R11	Skandha et. al. 2020 [62]	Automatic Features	Optimized CNN	2000 Images (346 Patients)	DL	95.66 *p* < 0.0001
R12	Saba et al. 2020 [63]	Automatic Features	CNN with 13 layers	2311 Images (346 Patients)	DL	89 *p* < 0.0001
R13	Proposed	Automatic Features	10 TL architectures VGG16 VGG19 DenseNet169 DenseNet121 XceptionNet MobileNet InceptionV3 AlexNet SqueezeNet ResNet50	346 Patients (Augmented from balanced to 6x)	DL	96.18 0.961 *p* < 0.0001
R14	Proposed	Automatic Features	SuriNet	346 Patients (Augmented from balanced to 6x)	DL	92.7 0.927 *p* < 0.0001

**Table 8 diagnostics-11-02109-t008:** Comparison of TL models.

SN#	Author, Year	Name of the Network	Dataset	Purpose	Pretrained Weight Size (MB)	Type of Layers
1	Krizhevsky et al., 2012 [72]	AlexNet	ImageNet	Classification	244	Convolution, Max Pooling, FCN
2	Simonyan et al., 2015 [66]	VGG -16, 19	ImageNet	Object recognition	528, 549	Convolution, Max Pooling, FCN
3	Szegedy et al., 2015 [69]	InceptionV3	ImageNet	Object recognition	92	Convolution, Max Pooling, Inception, FCN
4	He et al., 2016 [70]	ResNet 50, 101, and 152	ImageNet, CIFAR	Fast optimization for extremely deep neural networks	98,171, 232	Convolution, Avg Pooling, Residual, FCN
5	Howard et al., 2017 [42]	MobileNet	ImageNet	Classification and segmentation in mobiles	16	Convolution, Depth-wise Convolution, Average Pooling, FCN
6	Chollet et al., 2017 [71]	XceptionNet	ImageNet, JFT	Modified depthwise separable convolution. Advancement of InceptionV3	88	Convolution, Separable Convolution, Max Pooling, Global Avg Pooling, FCN
7	Huang et al., 2018 [48]	DenseNet 121, 169, 201, and 264	CIFAR	Gradient problem, substantially reducing the number of parameters	33, 57, 80	Convolution, Max Pooling, Transition, Dense, FCN, Global Avg Pooling
8	Landola et al. 2017 [73]	SqueezeNet	ImageNet	Reducing the number of parameters, efficient working on edge devices	4.8	Convolution, Fire Module Max Pooling, FCN Global Avg Pooling

**Table 9 diagnostics-11-02109-t009:** Similarities and differences between the TL models.

Architecture	Key Findings	Similarities	Differences
AlexNet	First deep neural network using convolution.	All the models are pre-trained on ImageNet All models use convolution operation Every model uses a softmax activation function in the output layer and a ReLu activation function in intermediate layers. Every model loads the pretrained weights from the cloud/offline. Every model uses a network-based TL paradigm.	MobileNet is focused on solving the computer vision problems in edge devices Densenet is trained and tested on the CIFAR dataset where remaining models uses ImageNet. XceptionNet only uses the JFT dataset for pre-training. Except for Xception and MobileNet, all the other models use standardized convolution. Except for IV3 and Xception, all other models use depth-wise kernels.
SqueezeNet	It is developed to reduce the number of parameters required for AlexNet with the same accuracy. Effectively used for edge devices.		
VGG	Reducing the number of parameters in convolution and training time.		
InceptionV3	Effective object detection for solving variable size objects using kernels of different sizes in each layer.		
ResNet	Solving the vanishing gradient problem in the deep neural network using skip (shortcut) connections.		
MobileNet	The first model was developed for supporting tensor flow in edge devices using light-weighted tensor flow.		
XceptionNet	Fast optimization and reducing the trainable parameters in IV3 using depth-wise convolution.		
DenseNet	Increasing the feed-forward nature in the neural networks using dense layers by concatenating the features from its previous layers.		

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
