# Peer review of "Ten Fast Transfer Learning Models for Carotid Ultrasound Plaque Tissue Characterization in Augmentation Framework Embedded with Heatmaps for Stroke Risk Stratification"

_diagnostics, 2021, doi:10.3390/diagnostics11112109_

Round 1

Reviewer 1 Report

This manuscript presents a novel approach to characterize plaque tissue within carotid arteries by using ultrasound images and transfer and deep learning approaches. The paper is very thorough and well structured.

The introduction section provides a very good overview of other findings in literature. The methods section describes in great detail all the used transfer learning and deep learning techniques. The results are clearly presented and discussed. The discussion section is very well written, with tables comparing different models proposed in literature and the obtained accuracies. There is also an analysis of similarities and differences among these models. At the end of Section 7, there is a good analysis of possible drawbacks of the presented study, and possible future improvements are also adequately discussed.

There are some corrections that will improve the quality of the manuscript.

Paragraph between lines 90 and 102 – the online and offline modules are discussed, but they are not described clearly enough. This segment should be rephrased, perhaps by referring the reader to Fig. 3, where the idea of online and offline system is well presented.

Section 2 is a little too detailed, since a similar overview of literature was also provided in the discussion section. Hence this section could be shortened.

The numbering of Figures and Tables, as well as their references in text, should be carefully checked. Figure 9 is mentioned before Figure 8, there are two Figures numbered with 11, Figure 10 mentioned in Line 389 is not in relation with SuriNet. The reader is referred to Table 7 in Line 549, while it should probably be Table 8 or 9.

Some abbreviations are mentioned without introduction, or are introduced later. Per example, the AUC is introduced in Line 412, while mentioned already in Line 353; abbreviation HOS was not introduced in text, only in the abstract.

Headline 5.3 seems unfinished.

The asterick in Table 4 is placed next to CNN11, while the explained abbreviations are valid for the entire Table, not just this row.

Figures, Tables and Equations in the Appendix section should have separate numbering.

The sentence in Line 548 should be rephrased.

The manuscript presents new techniques to analyze the type of plaque from ultrasound images of carotid arteries and offers the possibility to predict whether the plaque is symptomatic or asymptomatic. The good accuracy that was obtained proves that the presented approach has the potential of being a useful tool for the clinicians. There are some corrections that are mentioned in this Review that would additionally improve the manuscript. I think that with these corrections this manuscript can be accepted for publication.

Author Response

Reviewer-1

R1.1) This manuscript presents a novel approach to characterize plaque tissue within carotid arteries by using ultrasound images and transfer and deep learning approaches. The paper is very thorough and well structured.

Answer: Thanks for the encouragement. We appreciate the feedback from this reviewer. Thank you.

R1.2) The introduction section provides a very good overview of other findings in the literature. The methods section describes in detail all the used transfer learning and deep learning techniques. The results are presented and discussed. The discussion section is very well written, with tables comparing different models proposed in the literature and the obtained accuracies. There is also an analysis of similarities and differences among these models. At the end of Section 7, there is a good analysis of possible drawbacks of the presented study, and possible future improvements are also adequately discussed.

Answer: Thanks for your thorough study and encouragement. We appreciate the feedback from this reviewer. Thank you.

Some corrections will improve the quality of the manuscript.

R1.3) Paragraph between lines 90 and 102 – the online and offline modules are discussed, but they are not described clearly enough. This segment should be rephrased, perhaps by referring the reader to Fig. 3, where the idea of the online and offline system is well presented.

Answer: Thanks for allowing enhancing the quality of the manuscript. We rephrased the paragraph between 90 to 102 lines as follows:

The architecture of the proposed global AI model is shown in Figure 1. It contains five blocks (i) image acquisition, (ii) pre-processing, (iii) AI-based models, and (iv-v) performance evaluation and validation. The image acquisition block is used for scanning the internal carotid artery. These scans are normalized and manually delineated in the pre-processing block to obtain the plaque region-of-interest (ROI). Since the cohort size was small, we, therefore, added the augmentation block as part of the pre-processing step. The AI model block helps to determine the unknown label of the symptomatic vs. asymptomatic plaques. This is accomplished by transforming the test plaque image by the trained TL/DL models.  In our proposed framework, since there are 11 models, we run each test patient’s plaque using 11 (10 TL + 1 DL) different AI models for predicting 11 kinds of labels. We determine the performance of these 11 architectures and are followed by the ranking of their performance.”

Figure 1. Online AI architecture of the Atheromatic™ 2.0TL study (TL: Transfer Learning, DL: Deep Learning, and Grad-CAM: Gradient-weighted Class Activation Mapping).

R1.4) Section 2 is a little too detailed, since a similar overview of literature was also provided in the discussion section. Hence this section could be shortened.

Answer: Thanks for the question, we shortened section 2 on page number 7.  This is shown in the manuscript as follows:

The existing work on carotid plaque characterization using ultrasound with AI techniques is primarily focused on the machine learning paradigm. A handful of the studies are focused on using DL. Our study is the first of its kind that uses the TL paradigm embedded with heatmaps for PTC. The section briefly presents the works on PTC. Detailed tabulation is described in the discussion section. 

Seabra et al. [45] used graph cut techniques for the characterization of 3D ultrasound. It allows for the detection and quantification of the vulnerable plaque. The same set of authors in [46] estimated volume inside the ROI plaque using the Bayesian technique. They compared the proposed method with a gold standard and achieved better results with GSM<32. In [47], they characterized the plaque components such as lipids, fibrotic, and calcified using the Rayleigh mixture model (RMM).

Afonso et al. [48] proposed a CAD tool (AtheroRisk™, AtheroPoint, Roseville, CA, USA) to characterize the plaque echogenicity using activity index and enhanced activity index (EAI). The authors achieved an AUC of 64.96, 73.29, and 90.57% for the degree of stenosis, activity index, and enhanced activity index. This AtheroRisk™ CAD system was able to measure the plaque rupture risk. Loizou et al. identified and segmented the carotid plaque in M-mode ultrasound videos (MUV) using a snake algorithm [49-51]. In [52], the authors studied the variations of the texture features such as spatial gray level dependence matrices (SGLD) and gray level difference statistic (GLDS) in the MUV framework to classify them using an SVM classifier. Doonan et al. [53]  studied the relationship between textural and echo density features of carotid plaque by applying the PCA-based feature selection technique.  The authors showed a moderate coefficient of correlation (r) between these two features which have a range from 0.211 to 0.641. In addition to the above studies Acharya et al. [61] [82] [83], Gastounioti et. al. [81], Skandha et. al. [58], Saba et al [59] also conducted studies in the area of PTC using AI methods. This will be discussed in detail in section 5, labeled benchmarking

Thank you for the inputs in improving the readability of the manuscript.

R1.5)  The numbering of Figures and Tables, as well as their references in text, should be carefully checked. Figure 9 is mentioned before Figure 8, there are two Figures numbered with 11, Figure 10 mentioned in Line 389 is not in relation with SuriNet. The reader is referred to Table 7 in Line 549, while it should probably be Table 8 or 9.

Answer: Thanks for the question, we apologize for the figure numbering error from figure numbers 8 to 18, and table number referring to line 549. The following changes are made for figure numbering:

SN#

Page #

Previous Faulty Numbering

Corrected Current Numbering

1

13, 14

Figure 9

Figure 8

2

14

Figure 8

Figure 9

3

16

Figure 11

Figure 12

4

18, 19

Figure 12

Figure 13

5

20

Figure 12

Figure 14

6

22

Figure 13

Figure 15

7

23

Figure 14

Figure 16

8

23

Figure 15

Figure 17

9

23

Figure 16

Figure 18

10

24

Figure 14

Figure 16

11

24

Figure 16

Figure 18

12

25

Figure 15

Figure 17

13

26

Figure 16

Figure 18

14

31

Figure 15

Figure 19

The table numbers are in an orderly fashion and all the numbers are fixed and match the text information in the main body of the manuscript.

Thank you very much for your kind inputs and support in improving the readability of the manuscript. Thank you.

R1.6) Some abbreviations are mentioned without introduction or are introduced later. Per example, the AUC is introduced in Line 412, while mentioned already in Line 353; abbreviation HOS was not introduced in text, only in the abstract.

Answer: Thanks for giving the opportunity, we carefully examined the paper and have now added the abbreviations at the right place, which means, it appears for the first time in the revised manuscript. In addition to that, we added an abbreviation table at the beginning of the manuscript. Thank you much for helping improve the readability of the manuscript.

R1.7) Headline 5.3 seems unfinished.

Answer: Thanks for giving the opportunity, we changed the title of subsection 5.3 from “ Benchmarking the performance with existing” to “AUC-ROC Analysis”.

R1.8) The asterick in Table 4 is placed next to CNN11, while the explained abbreviations are valid for the entire Table, not just this row.

Answer: Thanks for giving me the opportunity, we updated the table 4 description as shown below. We have further updated the revised manuscript on page number 22. Thank you.

Table 4. Ranking table of the AI models.

Rank

Model

O

A

F

F1

Se

Sp

DS

D

TT

Me

AUC

AS

%

1

VGG19

5

3

4

5

5

4

5

5

3

1

3

43

78.18

2

MobileNet

2

5

4

3

5

4

1

4

5

5

5

43

78.18

3

CNN11*

4

5

2

4

5

4

4

5

1

3

5

42

76.36

4

AlexNet

5

4

2

2

2

2

5

3

4

3

3

35

63.60

5

Inception

1

3

5

5

4

5

1

5

1

1

3

34

61.82

6

DenseNet169

1

5

4

3

3

4

1

3

2

3

5

34

61.82

7

XceptionNet

5

3

2

2

3

2

5

0

3

4

3

32

58.18

8

SuriNet

2

3

3

3

4

3

3

3

3

3

30

54.55

9

VGG16

5

1

3

3

3

3

5

1

4

1

1

30

54.55

10

SqueezeNet

2

2

3

3

3

3

4

1

2

3

2

28

50.90

   11

DenseNet 121

4

2

2

2

3

2

4

0

2

3

2

26

47.27

12

ResNet50

3

2

2

2

3

2

3

0

1

3

2

23

41.80

O: Optimization, A: Accuracy, F: False Positive Rate, F1: F1 Score, Se: Sensitivity, Sp: Specificity, DS: Data Size, D: DOR, TT: Training Time, Me: Memory, AUC: Area-Under-the-Curve, and AS: Absolute Score. Note that CNN11 (rank 3) was used for benchmarking against other models (1,2, and 4-12).

R1.9) Figures, Tables, and Equations in the Appendix section should have separate numbering. The sentence in Line 548 should be rephrased.

Answer: Thanks for allowing increasing the quality of the manuscript, we have separately given the numbering for Appendix A, and B, as per your suggestions. This can be seen from pages # 39 to page # 41 in the revised manuscript. Thank you.

R1.10) The manuscript presents new techniques to analyze the type of plaque from ultrasound images of carotid arteries and offers the possibility to predict whether the plaque is symptomatic or asymptomatic. The good accuracy that was obtained proves that the presented approach has the potential of being a useful tool for the clinicians. There are some corrections that are mentioned in this Review that would additionally improve the manuscript. I think that with these corrections this manuscript can be accepted for publication.

Answer: Thank you very much for giving me the opportunity and encouragement. We appreciate it. Thank you once again.

Reviewer 2 Report

Comments:

  1. Title, “Ten Fast Transfer Learning Models for Carotid Ultrasound Plaque Tissue Characterization in Augmentation Framework embedded with Heatmaps for Stroke Risk Stratification”, please remove “Ten Fast Transfer Learning Models “ because the author mentions in the abstract at line 23 “ We applied 11 kinds of artificial intelligence (AI) models”. Suggestion: create a nice title without any number, e.g., “Transfer Learning based models for Carotid Ultrasound Plaque Tissue Characterization in Augmentation Framework embedded with Heatmaps for Stroke Risk Stratification”.
  2. The word “superior” in line 23 is not a good word in research. Please change it.
  3. In line 23, “We applied 11 kinds of artificial intelligence (AI) models, 10 of them were augmented and optimized using TL approaches”, why 10 models only augmented and optimized? What about the other one?
  4. In line 24, “a class of Atheromatic™ 2.0TL (AtheroPoint™, Roseville, CA, USA) that consisted of”, what does it mean? not very clear?
  5. In line 27, "Squeeze Net" should be SqueezeNet. Please check all the model names carefully and consist all in the same manner.
  6. Please consider for citing recent two good articles for skin cancer at line 67”, skin cancer [22, 23]”,
  • “Deep Learning in Medical Applications: Lesion Segmentation in Skin Cancer Images Using Modified and Improved Encoder-Decoder Architecture”, R Kaur, H GholamHosseini, R Sinha - Geometry and Vision, 2021.
  • “SLSNet: Skin lesion segmentation using a lightweight generative adversarial network”, MMK Sarker, HA Rashwan, F Akram, VK Singh- Expert Systems with Applications, 2021.
  1. Please also consider for citing recent two good articles for COVID pneumonia at line 84”, COVID pneumonia [42, 43]”,
  • “Web-based efficient dual attention networks to detect COVID-19 from X-ray images”, MMK Sarker, Y Makhlouf, SF Banu, S Chambon- Electronics Letters, 2020.
  • “COVID-19: Automatic detection from X-ray images by utilizing deep learning methods”, B Nigam, A Nigam, R Jain, S Dodia, N Arora - Expert Systems with Applications, 2021.
  1. Please improve the Figure 4 quality.
  2. Please improve all Figure text quality. Most of them are hard to read.
  3. Finally, there is some formatting and typographical error which need to be corrected.

Author Response

Reviewer-2

R2.1) Title, “Ten Fast Transfer Learning Models for Carotid Ultrasound Plaque Tissue Characterization in Augmentation Framework embedded with Heatmaps for Stroke Risk Stratification”, please remove “Ten Fast Transfer Learning Models “ because the author mentions in the abstract at line 23 “ We applied 11 kinds of artificial intelligence (AI) models”. Suggestion: create a nice title without any number, e.g., “Transfer Learning based models for Carotid Ultrasound Plaque Tissue Characterization in Augmentation Framework embedded with Heatmaps for Stroke Risk Stratification”.

Answer: Thanks for the giving opportunity and suggestion We applied 11 AI models out of which ten were transfer learning and one was deep learning model. We are sorry about the misunderstanding.  Since we are using 10 TL models, the title needs to be kept the same. Further, the  DL model is used for benchmarking. The main innovation of this study is the design and implementation of 10 TL models.

R2.2) The word “superior” in line 23 is not a good word in research. Please change it.

Answer: Thanks for allowing enhancing the quality of the manuscript, we used “improved performance” instead of “superior” at line 23.

R2.3) In line 23, “We applied 11 kinds of artificial intelligence (AI) models, 10 of them were augmented and optimized using TL approaches”, why 10 models only augmented and optimized? What about the other one?

Answer: Thanks for the question and we apologize for the confusion.

We applied 11 kinds of artificial intelligence (AI) models (10 TL and 1 DL), and all of them were augmented and optimized. These 11 models were (i-ii) Visual Geometric Group-16, 19 (VGG16, 19); (iii) Inception V3 (IV3); (iv-v) DenseNet121, 169; (vi) XceptionNet; (vii) ResNet50; (viii) MobileNet; and (ix) AlexNet, and (x) Squeeze Net, and one DL-based (xi) SuriNet-derived from UNet. We benchmark these11 AI models against our earlier generation consisting of Deep Convolutional Neural Network (DCNN), called Atheromatic™ 2.0DL.

I hope this clarifies the misunderstanding. Thank you for the opportunity.

In lieu of the above discussion, we have updated the abstract accordingly. Thank you again for giving us the chance to improve the readability of the manuscript.

R2.4) In line 24, “a class of Atheromatic™ 2.0TL (AtheroPoint™, Roseville, CA, USA) that consisted of”, what does it mean? not very clear?

Answer: Thanks for the question, Atheromatic™ 2.0TL is the name of the computer-aided diagnosis (CAD) system it contains ten transfer learning models and one deep learning model. The atheromatic system is a system for tissue characterization of the ultrasound vascular data. This is the second-generation system since our first-generation system was based on machine learning and was called Atheromatic™ 1.0 ML. So, the current system is a class of Atheromatic systems and the generation is 2.0. Thank you for understanding. We appreciate the inputs.

R2.5) In line 27, "Squeeze Net" should be SqueezeNet. Please check all the model names carefully and consist all in the same manner.

Answer: Thanks for the question, we updated the text at line 27  from “Squeeze Net" to ” SqueezeNet”. We carefully examined all the network names.

R2.6) Please consider for citing recent two good articles for skin cancer at line 67”, skin cancer [22, 23]”,

  • “Deep Learning in Medical Applications: Lesion Segmentation in Skin Cancer Images Using Modified and Improved Encoder-Decoder Architecture”, R Kaur, H GholamHosseini, R Sinha - Geometry, and Vision, 2021.
  • “SLSNet: Skin lesion segmentation using a lightweight generative adversarial network”, MMK Sarker, HA Rashwan, F Akram, VK Singh- Expert Systems with Applications, 2021.

Answer: Thanks for providing the opportunity, we cited the specified papers at line 67.

R2.7) Please also consider citing recent two good articles for COVID pneumonia at line 84”, COVID pneumonia [42, 43]”,

  • “Web-based efficient dual attention networks to detect COVID-19 from X-ray images”, MMK Sarker, Y Makhlouf, SF Banu, S Chambon- Electronics Letters, 2020.
  • “COVID-19: Automatic detection from X-ray images by utilizing deep learning methods”, B Nigam, A Nigam, R Jain, S Dodia, N Arora - Expert Systems with Applications, 2021.

Answer: Thanks for providing the opportunity, we cited the specified papers at line 67

R2.8) Please improve the Figure 4 quality.

Answer: Thanks for providing the opportunity, we improved the quality of figure 4. The improved figure was represented as follows

Figure 4. VGG16 and VGG19 architectures, CONV: Convolution Layer, FC: fully connected network.

R2.9) Please improve all Figure text quality. Most of them are hard to read.

Answer: Thanks for providing the opportunity, we improved the quality of figures and their readability. We found that except for figure 15 all the figures are better readable. We updated figure 15 (ROC) which is represented as follows

Figure 15. ROC comparison of 12 AI models (10 TL, 1 DL-SuriNet, and 1 DL-DCNN).

R2.10) Finally, there is some formatting and typographical error which need to be corrected.

Answer: Thanks for providing the opportunity, we carefully formatted the manuscript, and corrected typographical errors related to figures and tables.